# Development of a Flash Flood Confidence Index from Disaster Reports and Geophysical Susceptibility

**Andrew Kruczkiewicz** [1,2,3,*], **Agathe Bucherie** [1], **Fernanda Ayala** [4], **Carolynne Hultquist** [5], **Humberto Vergara** [6,7], **Simon Mason** [1], **Juan Bazo** [2,8] **and Alex de Sherbinin** [5]

1   International Research Institute for Climate and Society (IRI), The Earth Institute at Columbia University, Palisades, NY 10964, USA; agathe@iri.columbia.edu (A.B.); simon@iri.columbia.edu (S.M.)
2   Red Cross Red Crescent Climate Centre, 2953 The Hague, HT, The Netherlands; bazo@climatecentre.org
3   Faculty of Geo-Information Science and Earth Observation, University of Twente, 7514 Enschede, AE, The Netherlands
4   Disaster Risk Department, Ecuadorian Red Cross (CRE), Quito 170403, Ecuador; mayala@cruzroja.org.ec
5   Center for International Earth Science Information Network (CIESIN), The Earth Institute at Columbia University, Palisades, NY 10964, USA; c.hultquist@columbia.edu (C.H.); adesherbinin@ciesin.columbia.edu (A.d.S.)
6   Cooperative Institute for Mesoscale Meteorological Studies (CIMMS), The University of Oklahoma, Norman, OK 73019, USA; humber@ou.edu
7   NOAA National Severe Storms Laboratory (NSSL), Norman, OK 73072, USA
8   Universidad Tecnológica de Peru (UTP), Lima 15046, Peru
*   Correspondence: andrewk@iri.columbia.edu

**Abstract:** The analysis of historical disaster events is a critical step towards understanding current risk levels and changes in disaster risk over time. Disaster databases are potentially useful tools for exploring trends, however, criteria for inclusion of events and for associated descriptive characteristics is not standardized. For example, some databases include only primary disaster types, such as 'flood', while others include subtypes, such as 'coastal flood' and 'flash flood'. Here we outline a method to identify candidate events for assignment of a specific disaster subtype—namely, 'flash floods'—from the corresponding primary disaster type—namely, 'flood'. Geophysical data, including variables derived from remote sensing, are integrated to develop an enhanced flash flood confidence index, consisting of both a flash flood confidence index based on text mining of disaster reports and a flash flood susceptibility index from remote sensing derived geophysical data. This method was applied to a historical flood event dataset covering Ecuador. Results indicate the potential value of disaggregating events labeled as a primary disaster type into events of a particular subtype. The outputs are potentially useful for disaster risk reduction and vulnerability assessment if appropriately evaluated for fitness of use.

**Keywords:** flash flood; disaster risk reduction; historical disaster database; flood characterization; geomorphology; geospatial analysis; Ecuador; disaster management; text analytics; early warning system; climate informed decision making; flood risk

## 1. Introduction

Weather-related disaster frequency and impacts are increasing globally due to climate and human behavior changes [1,2]. Amongst flood types, flash floods are the most deadly, causing more than 5000 fatalities worldwide annually [3,4]. While there is a lack of a universally accepted definition of flash flood [5,6], the term most commonly refers to rapid onset hydrological events triggered by localized, intense rainfall events, resulting in unexpected, sudden and high velocity flow occurring in small streams or artificial waterways, as well as surface runoff [7]. However, the term 'flash flood' can also refer to events related to the collapse of sewage systems, river bank or dam failure, as well as any pluvial flood events resulting in surface run-off happening far from the river network, in

urban or rural environments [5,8]. Aggravated by steep topography, flash floods can modify river courses, overtake livestock, propagate debris, and destroy or bury infrastructure with sediment [4]. Flash floods are more challenging to monitor, document, characterize and predict than other flood types due to their localized and sudden nature [9–12]. The lack of documentation of historical flash flood events and their impacts, at sufficient levels of granularity, has contributed to less predictability and limited the development of services and programs to assess flash flood vulnerability and exposure [6,13,14]. This paper addresses how combining records of historical flood event with remote sensing inputs enriches our understanding and assessment of flash flood risk.

### 1.1. Historical Data for Disaster Events

Historical data that includes disaster magnitude, damage and extent—produced either by disaster practitioners, media or remote sensing reports—are necessary to understand if and to what extent risk for a specific disaster type (and subtype) is present in an area of interest. Without historical data, under- and over-estimating both the static representations of risk [15–17], as well as the potential impacts from a specific event, may occur [18–20]. Historical records of high impact and infrequent disaster types, such as flash floods, are of particular value to improve risk assessment and predictability [21–23]. However, in many locations, historical disaster data have insufficient spatiotemporal resolution for assessment of risk [23].

Remote sensing is increasingly used for disaster monitoring and characterization at a high spatial resolution [24–26]. However, some disaster subtypes, such as flash floods, are not often captured by satellite monitoring systems [14,27,28]. To develop a climatology for a specific disaster subtype, it is necessary to use datasets that include data over a sufficient temporal range [29], at an appropriate level of granularity [30,31], and with a significant sample size [32]. Multiple databases may lead to improved outputs, however, the quality and fitness for use of each must be assessed [33,34] as inconsistencies may exist, such as variations in standard operating procedures for event entry, in the description of detail and differences in criteria for aggregation of events into a primary type and disaggregation into subtypes [35,36]. For example, frequency of occurrence, magnitude and seasonality will likely vary across databases [37], however, standards in integration, clearing and merging data will contribute to more accessible products [38]. Remote sensing has shown value in corroborating and enhancing historical disaster datasets [39], especially for specific types of floods such as riverine [40], storm surge [41] and glacial lake outburst floods [42], however, progress related to flash floods lags behind.

### 1.2. Flash Flood as a Disaster Subtype

Disaggregation of flood type is particularly important in geographic areas where numerous types of floods can occur [43,44]. While some flash flood specific datasets exist (such as HYDRATE and HANZE in Europe, FLASH in USA), spatial and temporal coverage is limited [45,46]. In databases with records of floods, it is uncommon for criteria on event inclusion to require a declaration of flood subtype [47–49]. Even in databases that include subtypes, protocols are unlikely to exist for disaggregating, tagging [50,51] or for attributing magnitude of direct and indirect impacts [52]. However, in a database including only primary flood type, there are likely to be some attributes that allow for the assignment of a subtype (such as 'flash flood') to some degree of confidence [53].

Appropriate and acceptable anticipatory action differs per flood type given the difference in their impact profile [14,54,55]. For example, with a 5-day lead time, an effective anticipatory action for flash floods could include advanced clearing of sewage systems [56,57], although this would not likely be a priority anticipatory action for riverine floods due to the lower importance of sewage-system conditions related to riverine flood risk. While potentially justified as a method for simplification, combining flood subtypes in historical databases can result in the loss of critical data that could lead to the dataset lacking the specificity needed for meaningful inference related to flood type specific risk [58–61].

These uncertainties and lack of specificity in flood type reporting propagate through to applications of derived datasets, tools and services, such as those intended for disaster resilience and risk reduction programs, and damage calculation [62].

### 1.3. An Index for Flood Type Classification

Various methods exist to differentiate between disaster types within a dataset, however, most consist of an ad hoc review for type-specific terms and phrases within an event report [63,64]. This highlights the importance of identifying terms and phrases that are representative of the subtype of interest [65]. Relative to enhancing disaster data, index-based approaches have been employed to extract certain descriptors of disaster, such as the sentiment of the affected populations and impact magnitude [66–68]. While historical data for events noted only as 'flood' exists, a widely accepted approach to assign a flood subtype to these events does not [69]. We acknowledge that assigning flood type will likely be imperfect for some events, however, given the demand for historical data on flash floods, we argue that using an index-based approach could provide a sufficient level of certainty in event subtype classification.

Here, we developed an index to fill the gap in historical flash flood content using a data mining approach. We calculated a numeric value representing the confidence that an event is a flash flood based on its event description and the geophysical properties of the location. The primary objective of this flash flood confidence index is to enhance flash flood risk analyses by allowing users to extract likely flash flood events from datasets that contain flood events labelled only as 'flood'. The resultant subset of likely flash flood events could be a valuable tool in developing flash flood specific risk assessments, climate services and early warning systems. Here we describe a use case for developing anticipatory humanitarian action protocols, however, the approach is designed to be applied to other risk and vulnerability assessment processes.

### 2. Materials and Methods

Using remote sensing-derived geophysical data and descriptive records of disaster events, we produced a subset of events that are likely to be flash floods. In this section, a three-step method is outlined: (1) identification and compilation of historical data; (2) development of a flash flood confidence index using identifiers from descriptive event records; and (3) enhancement of the confidence index using event location and extracted geophysical data into a flash flood susceptibility index.

### 2.1. Step 1—Identification and Compilation of Historical Data
#### 2.1.1. Text-Based Data

Disaster reports can contain information on disaster type and impact critical for further characterization of the event by subtype [70]. Examples of types of sources include [71]: government (local and national); non-governmental (NGO) and humanitarian; private sector and academic (including journal publications); and media (such as news reports, crowdsourcing, and social media) [72–74]. Historical flood data are globally available in various databases [75] such as DesInventar, the International Federation of Red Cross Red Crescent Societies Go Platform, Dartmouth Flood Observatory [76] and the Emergency Events Database (EM-DAT) [77]. However, these sources can have substantial inhomogeneities in spatial coverage, temporal range, and disaggregation of type and subtype [78]. Therefore, to compile data into a derived historical dataset, preprocessing is necessary, including, but not limited to, collation and cleaning [6]. Doing so increases quality, comprehensiveness and consistency [79] and decreases the risk for event duplication [80]. If available, a geographic feature, such as centroid or polygon of extent for each event should be included in the 'cleaned' dataset. If geographic information does not exist within the original record, geographic attribute data can be potentially manually assigned based on location-specific geographical terms, such as community name or roadway, and expert judgement [81,82].

### 2.1.2. Geophysical Data

Various remote-sensing-derived datasets are available globally at sufficient resolution to support flash flood susceptibility analyses. Examples of Earth Observation data that inform flash flood susceptibility are described in Section 2.3, including Digital Elevation Models (DEMs) from NASA's Shuttle Radar Topography Mission (SRTM), which are used to measure hypsometric variables such as steepness of slope and terrain curvature [83].

### 2.2. Step 2—Development of a Flash Flood Confidence Index (FFCI)

Here, we created an index to assess the likelihood of a primary-level event (flood) report having the characteristics of the subtype of interest (flash flood). We first defined categories (meteorological, hydrological, environmental, and flood dynamic) of flash flood drivers and characteristics based on previous research and expert knowledge on flash flood processes and definition [5,6,8,84]. Primary categories (Table 1, column 1) were organized into subcategories, such as 'Heavy precipitation' in 'Meteorological' and 'Strong current' in 'Flood dynamics' that are most commonly associated with flash flooding [85–87]. Local flood and disaster experts participated in keyword selection for each subcategory (such as 'strong' and 'intense' in 'Heavy precipitation'), as these lists of terms were used for mining the historical report description (Table 1, column 2).

**Table 1.** Categories and subcategories of flash flood drivers. Flash flood keywords and score indicating a subcategory relative contribution to flash flood risk.

| Categories (and Subcategories) | Keywords | Score |
|---|---|---|
| A- Meteorological | | |
| A.1- Heavy precipitation | Strong, intense, heavy precipitation and rainfall, stormy rain | 2 |
| A.2- Short duration precipitation | Precipitation of a few hours, occurring at a specific moment of the day (morning, afternoon, evening, night … ) | 3 |
| B- Hydrological | | |
| B.1- Small stream overflow | Overflowing of streams, ravines, gullies | 4 |
| B.2- Artificial waterways overflow | Ditch, canals, Irrigation channels, gutters, artificial waterways | 1 |
| B.3- River bank failure | Failure, rupture of river banks | 4 |
| B.4- Surface runoff | Mudflow, superficial, surface water, running water | 6 |
| C- Environmental | | |
| C.1- Urban | Flooded streets, blocked drainage systems, sewage systems issues | 1 |
| C.2- Slope | Water coming from the hills, steep terrain | 5 |
| D- Flood dynamics | | |
| D.1- Strong current | Strong, fast moving water, torrential currents, dragging force, sweeping away, collapsing walls, destructive, dangerous events | 7 |
| D.2- Short-duration flood | Flash, unexpected, water level quickly returning to normal, water evacuated within a few hours | 7 |

The 10 sub-categories were ranked based on their relevance to the flash flood definition and the frequency of occurrence in the report description [70,73]. As an example, nearly all flash floods are triggered by heavy precipitation, but not all heavy precipitation events trigger a flash flood, explaining the relative lower score for Subclass A.1. On the other hand, keywords related to strong currents (D.1) and short-duration (D.2) floods are more likely to represent flash flooding processes, and therefore, were assigned higher scores.

For a flood event report to be eligible for flash flood index scoring, it must first satisfy the requirement to have at least one keyword in the report description. If yes, the corresponding subcategories scores were summed, defining the event's FFCI value, from 0–40. The FFCI value is interpreted as a diagnostic measure of an event being a flash flood. For simplicity, the FFCI values were rescaled from 0 to 10, either using a limit at 10, or a classification method, depending on the resulting FFCI distribution specific to the input historical dataset.

*2.3. Step 3—Enhancement of the Index Using Location and Geophysical Susceptibility Data*

Using location data within reports and satellite-derived geophysical data, we developed an enhanced FFCI (eFFCI), and subsequently applied it to assess the degree to which an event is located in a polygon of relatively high or low flash flood susceptibility (Figure 1). The eFFCI should be applied only to reports with location data.

**Figure 1.** Schematic of eFFCI Architecture.

2.3.1. Flash-Food Susceptibility Index (FFSI) Development

Flash flood susceptibility assessments are driven by geospatial data [88,89] and primarily implemented at sub-national and basin scales, as exemplified in Vietnam [90], Iran [91,92], and Pakistan [93]. Hypsometry, drainage systems, and surface properties influence catchment hydrologic response to rainfall, and thus runoff generation and ultimately, flash flood risk [94–96]. Various factors drive the flash flood susceptibility of a catchment [93,97–99]; seven of the most common were selected for this approach. These factors were classified into three categories: hypsometry, drainage network, and surface properties [92,100–102] (see Table 2), and estimated or averaged over catchment areas (as delineated by HydroSHEDS). A flash flood susceptibility index (FFSI) represents the potential of a catchment to generate a flash flood when significant local rainfall occurs [103–106]. The FFSI was calculated at the catchment scale using a weighted mean of each indicator 'j', as presented in the workflow of Figure 2.

**Table 2.** Flash flood susceptibility indicators calculated at catchment scale, and related information for hypsometry, drainage network and surface properties.

| | Indicators | Data Source | Description |
|---|---|---|---|
| **Hypsometry** | Mean slope (deg) | Global SRTM 90 m Digital Elevation Model (DEM) v4.1 [107] | The mean slope is an indication of the flashiness of a watershed, proportional to the flood susceptibility [108]. The slope is computed in degrees from the DEM using 2nd degree polynomial adjustment algorithm [109]. |
| | Mean profile curvature (1/m) | | The profile curvature of a terrain is the rate of change of the slope gradient in the direction of steepest slope [110,111]. Negative profile curvatures (convex landforms), also related to erosion-dominated landscapes, are indicative of surface runoff and torrential flood. |
| **Drainage network** | Upslope contributing area $(km^2)$ | WWF HydroSHEDS v1 global datasets (15 arc-seconds resolution): level 12 hydrological basins [112] and river routing network datasets [113]. | The contributing area of a river point is correlated with its discharge potential [114]. The Upslope contributing area [112] is used to differentiate the upstream to more downstream catchment position within a country, and therefore guide the type of flood behavior to expect (from short onset flash floods to long onset riverine floods). |
| | Drainage density Dd $(km^{-1})$ | | The drainage density Dd $(Km^{-1})$ is a measure of the cumulative river length over the catchment area. This has a direct correlation with runoff potential, and therefore indirect correlation with infiltration rate [115,116]. |
| | Mean Strahler stream order | | The Strahler hierarchical river stream order [117], averaged across catchment, provides an indication of the basin mean stream order. A lower basin order corresponds to a higher proportion of small streams, and therefore higher flash flood potential [115,118]. |
| **Surface properties** | Sand content | ISRIC SoilGrids global dataset [119]. Sand fraction 250 m resolution product of the 0–5 cm depth. | Sand content is used as a proxy for infiltration potential of soils. The infiltration rate decreases with decreasing sand content, increasing runoff and flash flood susceptibility [88,120] |
| | Land use and land cover (LULC) | Copernicus Global Land Operations, derived from PROBA-V satellite observations, at 100 m [121]. | LULC directly impacts runoff generation and behavior [88,120,122]. Discrete LULC classes are reclassified into flash flood susceptibility scores from 1 to 8, depending on potential to influence surface runoff. Closed forest = score of 1, urban environment = score of 8. (see Appendix A for more details) |

Hypsometry characteristics, such as slope and curvature, are key elements in describing runoff acceleration of a catchment [108,110,123–125]. Terrain data were derived from NASA's Shuttle Radar Topography Mission (SRTM) 90 m Digital Elevation Model (DEM) v4.1 [107], and cleaned using a 2nd-degree polynomial adjustment algorithm [109]. Available globally, the 90 m SRTM DEM data were used to derive terrain variables [126,127].

Drainage network characteristics were extracted from the World Wildlife Fund (WWF) HydroSHEDS v1 global data, at a resolution of 15 arc-seconds. Level-12 hydrological basins [112,128] and river routing networks data were used [113]. HydroSHEDS' upslope basin contributing area was used to discern upstream (shorter response time) from catchments downstream (longer response time) [129–131]. Cumulative drainage density and mean drainage order were derived from the river-routing network, complementing the hypsometric variables [132,133].

Surface properties, such as vegetation cover, soil properties and land use, affect water infiltration and runoff behavior, and thus, they are considered here as relevant factors for flash flood risk assessment [88,94,134,135]. The Land Use Land Cover (LULC) product from Copernicus Global Land Operation [121] represents these surface properties over a variety of ecosystems (e.g., forests, grasslands, croplands, wetlands, urban areas), with the ISRIC SoilGrid Sand Fraction accounting for the infiltration potential of soils [119].

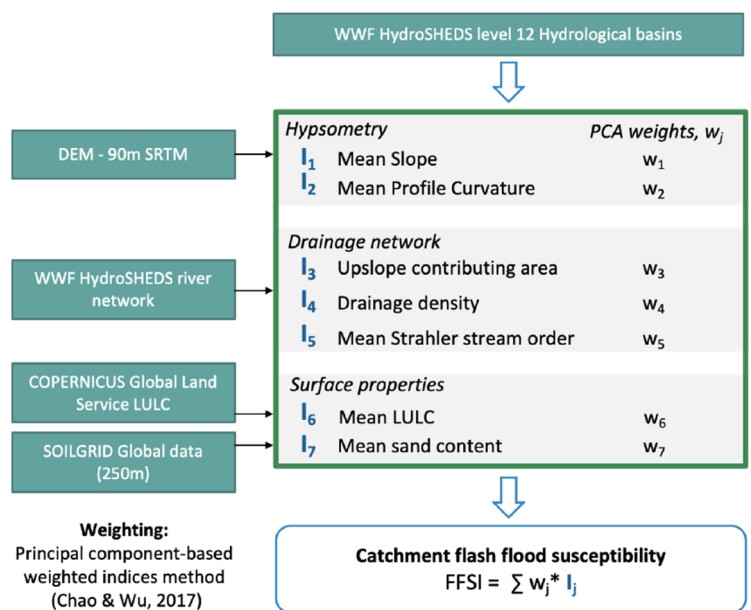

**Figure 2.** FFSI workflow for catchment scale, including remote sensing input data, geospatial indicators and resulting weights used in FFSI calculation.

Various weighting methods for constructing composite indicators exist [136], including equal weighting [137], weighting based on ranking of indicators [138], and variance-analysis methods [139]. Here we estimated the weight of each indicator ($w_j$) based on the results from a principal component analysis (PCA) [140]. Geomorphological indicators were normalized from 0 to 1, then set to the same direction of increasing susceptibility to flash flooding and standardized, resulting in a mean of 0 and a standard deviation of 1, before the calculation of the first principal component analysis scores [136,141]. A log transformation was then used to adjust potential skewed distributions prior to the PCA [142]. The weight of each indicator was obtained from its absolute variance proportion in the PCA equation following the principal component-based weighted indices method [140]. Next, the FFSI was attributed to each catchment (Equation (1)), resulting in a value from 0 to 1. The FFSI was then reclassified into discrete classes from 1 to 10 with the objective of matching the FFCI structure. Depending on the FFSI geographic context such as the spatial distribution of the input data, discretization methods using quantiles, equal intervals or standard deviations could be considered [143,144], as well as rule-based discretization specific to the case study.

### 2.3.2. Enhanced FFCI (eFFCI): FFCI Coupled with FFSI

After calculating a FFSI value for each event, the FFCI and FFSI values were combined using a weighted mean approach [145] to produce an Enhanced FFCI (eFFCI) described in Equation (1), where $\alpha$ represents the weight (from 0 to 1) assigned to FFCI relative to FFSI, to be defined. Equal weighting ($\alpha = 0.5$) methods are common, however, in doing so, a risk for under and over representation of variables should be considered [146,147].

$$\text{eFFCI}_i = \alpha * \text{FFCI}_i + (1 - \alpha) * \text{FFSI}_i \qquad (1)$$

with $\alpha = 1$ for events with no coordinate location.

## 3. Results

We developed a historical dataset of flood events that are likely to be flash floods, for continental Ecuador. Ecuador is located in northwestern South America from latitude 1.5° N to 5.0° S, and longitude 81.0° W to 75.0° W, excluding the Galapagos Islands, which are not included in this study. Transected longitudinally by the Andes mountains,

the geography of continental Ecuador is divided into three major natural regions; the Coastal, the Andes and the Amazonian regions [148], each characterized by different flood-risk dynamics [149]. West of the Andes, the coastal region features hills up to 800 m elevation and interior silted valleys. Ecuador has nine climatic regimes, according to the Köppen-Geiger classification [150], ranging from arid in some coastal areas, tropical in the lowlands to polar in the Andes. The spatial and temporal distribution of rainfall differs across regions [151], with a December–April rainy season on the coast and a longer season (September–May) in the Andes. The Amazon region receives rainfall year-round, with the highest amounts occurring in March–June [152]. Ecuador provides an ideal case study to differentiate flood subtypes due to its complex topography and flood patterns.

### 3.1. Building an Historical Dataset for Ecuador

Two databases (see Table 3) of historical disasters in Ecuador include information on individual flood-events, including location. The free and open DesInventar Database [153] from the United Nations Office for Disaster Risk Reduction (UNDRR) aggregates reports at administrative level 3 (Parroquia), since 2007. Servicio Nacional de Gestión de Riesgos y Emergencias (SNGRE), the National Risk and Emergency Management Service of Ecuador records impact and geographic coordinates since 2014. A four-step process was employed to clean and merge the two datasets: (i) merging into a common template; (ii) identifying duplicate events (same date and location); (iii) merging information for each SNGRE duplicate events (description, location, impact) into DesInventar data; and (iv) deleting the SNGRE duplicates. The derived dataset contains 3365 events, each with date, time, affected administrative level 3, location name and description of impact; 2194 include coordinate locations. The spatial distribution of the derived flood dataset of Ecuador is presented in Figure 3.

**Table 3.** Attributes of historical flood datasets in Ecuador and for the derived dataset.

| Dataset Name | DesInventar | SNGRE | Derived Dataset |
|---|---|---|---|
| Time range | 2007–2019 | 2014–2019 | 2007–2019 |
| Number of reports | 2859 | 2207 | 3365 |
| Spatial resolution characteristics | • Admin level 3<br>• no coordinate location | • Admin level 3<br>• 2194 reports with coordinate location | • Admin level 3<br>• 2194 reports with coordinate location |
| Flood and impact description | Yes | Yes | Yes |

### 3.2. Application of Flash Flood Confidence Index (FFCI) for Ecuador

The FFCI process was applied to the derived dataset of 3365 historical flood events. An event is categorized as a potential flash flood if the FFCI is non-0, with higher values indicating a higher likelihood of being a flash flood; 69.3% of flood events had at least one flash flood identifier (Table 4).

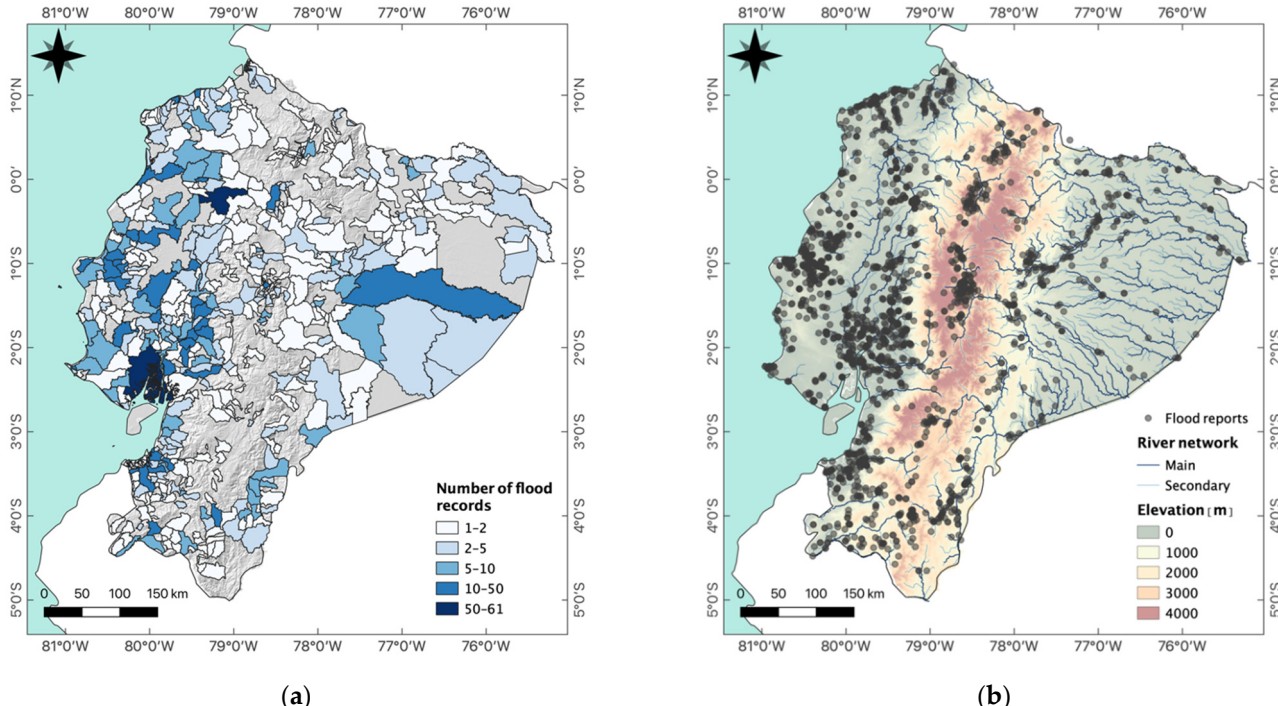

**Figure 3.** Historical flood events from the derived dataset (2007–2019): (**a**) Number of events per administrative level 3 (Parroquia) based on the 3365 historical records; (**b**) Location of the historical flood reports (2194) that included coordinates.

**Table 4.** Number of flash flood identifiers assigned per historical flood event.

| No. Identifiers | 0 | 1 | 2 | 3 | 4 | 5 | 6 |
|---|---|---|---|---|---|---|---|
| No. reports | 1033 | 1163 | 849 | 237 | 73 | 9 | 1 |

The distribution of FFCI for all flood events is presented in Table 5. Upon reviewing the distribution, critical thresholds can be set depending on user intent to identify either a quantity or percentage of events. This flexibility promotes transferability of the methods to a variety of use cases and geographic areas [154]. For example, if from expert interpretation a threshold at 5 is defined as critical for a specified use case, 1185 events are at or above the threshold of 5, representing 35.2% of all events and 50.8% of all non-0 events. However, in other use cases, a threshold of 5 may not be significant, therefore, this method is designed to allow for a user to set a critical threshold based on their perception of what is best: either a higher or lower percentage of events, or number of events [155,156].

**Table 5.** FFCI distribution historical flood events in Ecuador.

| FFCI | 0 | 1 | 2 | 3 | 4 | 5 | 6 | 7 | 8 | 9 | 10 |
|---|---|---|---|---|---|---|---|---|---|---|---|
| No. reports | 1033 | 155 | 596 | 259 | 137 | 385 | 180 | 158 | 72 | 141 | 249 |
| % => FFCI | – | 100 | 93 | 68 | 57 | 51 | 34 | 27 | 20 | 17 | 11 |

### 3.3. Application of Enhanced FFCI (eFFCI) for Ecuador
#### 3.3.1. FFSI

To administer the FFSI in Ecuador, the previously described (Section 2.3.1) principal component analysis was applied to assess the weighting of each indicator. The resulting weights are presented in Table 6. Proportionally, more weight was assigned to drainage network and hypsometry, consistent with the understanding of surface properties being less significant drivers of flash flood susceptibility than slope or drainage-basin size [100,157,158]. The composite FFSI map of Ecuador catchments, resulting from the

seven indicators is presented in Figure 4. The complete FFSI dataset for Ecuador, including the raw indicator values, as well as the normalized and classified FFSI results for each catchment are openly available on Zenodo [159].

**Table 6.** PCA-derived indicator weights.

| Categories | Hypsometry | | Drainage Network | | | Surface Properties | |
|---|---|---|---|---|---|---|---|
| Indicators (Ij) | 1 Mean slope | 2 Mean profile curvature | 3 Upslope contributing area | 4 Drainage density | 5 Mean Strahler stream order | 6 Mean LULC | 7 Sand content |
| PCA resulting weights (wj) | 0.18 | 0.10 | 0.18 | 0.20 | 0.14 | 0.12 | 0.08 |

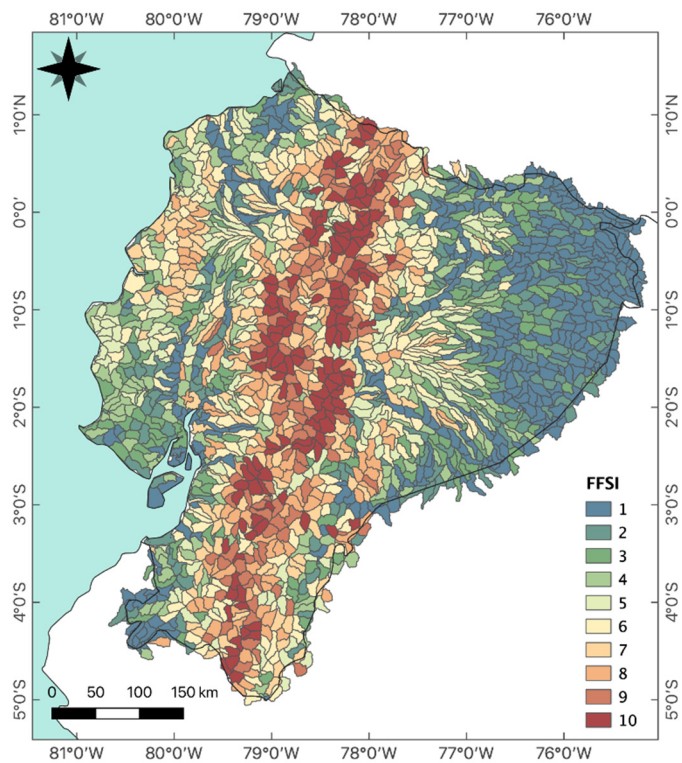

**Figure 4.** Result of the flash flood susceptibility index (FFSI) applied to continental Ecuador, as a composite map indicating relative susceptibility at the catchment level to flash flood.

### 3.3.2. eFFCI

To apply the eFFCI (Equation (1)) to the 2194 events with known locations, the parameter $\alpha$ (related to weighting for FFCI and FFSI) was estimated. The FFSI score was weighted higher for lower-confidence FFCI events, based on consultation with both disaster managers and scientists who are familiar with the context of flash floods in Ecuador, and are based on Ecuador and the region. To do so, the normalized FFCI values were assigned to the weight n Equation (2), and we used $(1 - \alpha)$ as a weighted factor for FFSI values. As FFSI values were uniform for all events occurring in the same catchment, the variability of flood type occurring within a catchment was addressed by this step.

$$\alpha = (\text{FFCI}_{max} - \text{FFCI}_i)/(\text{FFCI}_{max} - \text{FFCI}_{min}) \tag{2}$$

The eFFCI was then calculated for the 2332 historical events with at least one flash flood identifier. For the 753 events with no coordinates, n was assigned to 1 so that eFFCI = FFCI. For the remaining 1579 events with coordinates, eFFCI was calculated with

Equations (1) and (2). Table 7 shows the distribution of eFFCI results, indicating the percentage of events in the derived dataset that would be identified as 'likely flash flood events' for a given eFFCI threshold. For example, an eFFCI threshold of 5 would define 1369 events as 'most likely' to be flash floods (59% of the 2332 events). Figure 5 shows the count of FFCI and eFFCI events per score. eFFCI leads to an increase, compared to FFCI, in event count for higher score values (6–10), and a decrease for lower values (1–3).

**Table 7.** eFFCI distribution of Ecuador historical flood events.

| eFFCI | 0 | 1 | 2 | 3 | 4 | 5 | 6 | 7 | 8 | 9 | 10 |
|---|---|---|---|---|---|---|---|---|---|---|---|
| No. reports | - | 68 | 432 | 227 | 236 | 307 | 256 | 222 | 150 | 163 | 271 |
| % => eFFCI | - | 100 | 97 | 79 | 69 | 59 | 46 | 35 | 25 | 19 | 12 |

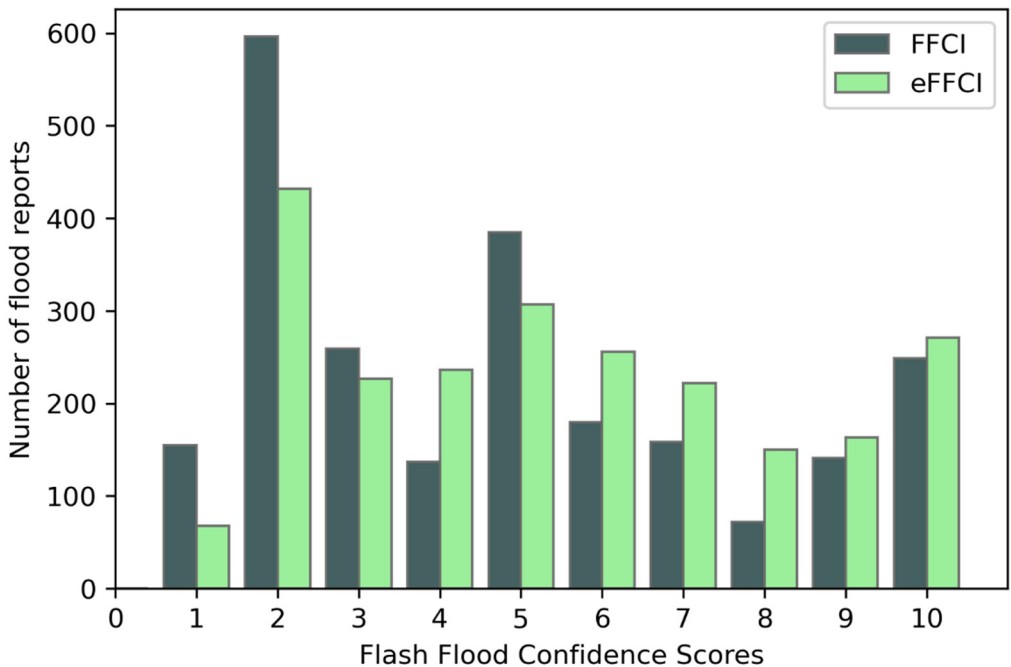

**Figure 5.** Distribution of FFCI and eFFCI applied to 2332 potential flash flood events.

## 4. Discussion

### 4.1. Interpretation of the FFCI

The FFCI values can be used as a discrete confidence index to analyze correlations between other parameters including hydro-meteorological (such as precipitation) and impact-related. The critical values within the FFCI that are used to define bins that relate to confidence levels must be independent of the application, however, the application context may lead to selection of different 'critical levels' of flash flood confidence. For example, in the humanitarian anticipatory action context, the tolerance to uncertainty may be lower than other use cases, and as such may necessitate a lower critical threshold to be selected. Alternatively, if the tolerance to uncertainty is higher for a particular use case, perhaps for applications beyond the humanitarian context such as in energy and financial sectors, the critical threshold could be raised [160,161].

In Figure 6, based on discussions with stakeholders in Ecuador, the critical threshold of 5 is defined to differentiate events that are 'more-likely' or 'less-likely' to be flash floods. At this level, the proportion of flash floods is higher in the Andes than the Coastal areas regardless of the month, with relative maxima for flash flood proportion in the Andes occurring in May, October, November and December (65%, 62%, 67%, 61%, respectively). However, the quantity of likely flash floods reports is higher in other months, such as April. In the Amazon, while the count is highest in April, the proportion of flash flood to

non-flash flood is highest in January and October (both equal to 52%), with February next at 50%. Further, the seasonality for total flood events, total flash flood (and non-flash flood) events and proportion of flash flood events is shown. The Coastal areas experiences the highest proportion of flash flood events in months where relatively low total counts of flood events occur (December, October, June). This pattern is also seen (albeit less prominently) in the Andes with 3 of the 4 months with a proportion of flash flood greater than 60% being months with low counts of total events (October, November, December). This potentially indicates how seasonality of flood-type likelihood can be integrated into the design of early warning and anticipatory action programming to include contingencies to prioritize (or de-prioritize) certain flood types, in certain regions, on a seasonal time scale [162]. Additional questions arise related to the degree to which areas that are considered to be at 'low risk' to 'flood' in certain seasons/months may actually be at a higher level of risk for a 'flash flood', relative to other months. This may be useful in developing uncertainty and risk communication programs as well as educational outreach strategies to build community resilience [163–166].

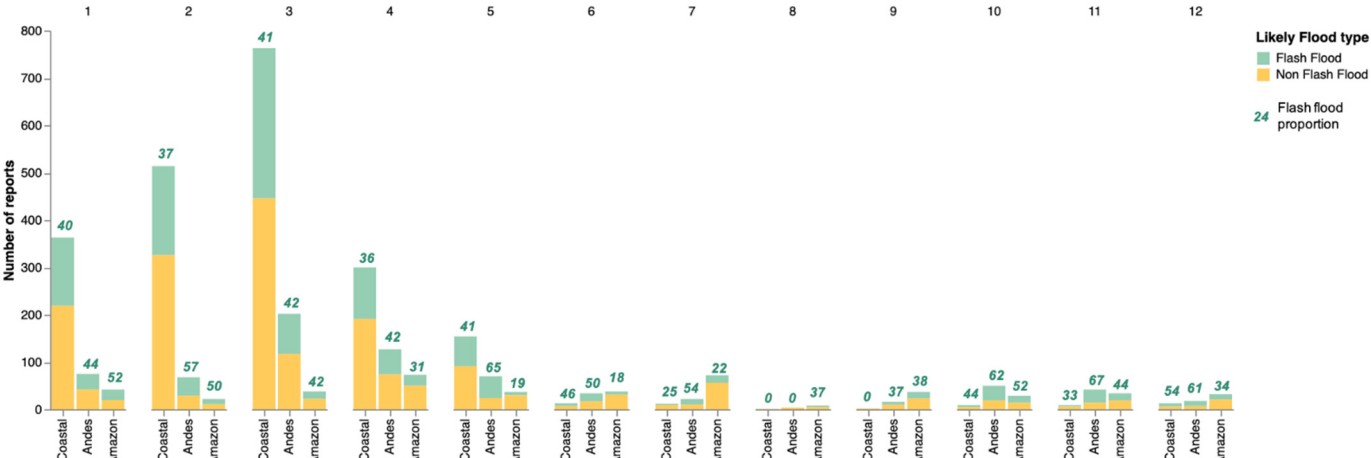

**Figure 6.** Count of flood events by month, for each region of Ecuador, and proportion of flash flood to non-flash flood events. Proportion of 'more likely' versus 'less likely' flash flood events based on an eFFCI threshold of 5.

### 4.2. Benefit of FFCI

The FFCI approach filters likely flash flood events from historical disaster databases that currently do not assign a flood subtype. If the FFCI-derived dataset of 'likely' flash floods did not exist, it would add considerable difficulty and require additional resources to delineate flash flood risk across Ecuador—a task that is a critical step in developing early warning and anticipatory action triggers and standard operating procedures for the Ecuador Red Cross and other organizations involved with disaster management [167]. Without this dataset, an anticipatory action program for flash floods would be potentially designed inappropriately as it would likely be conditioned on: a small sample size (the six 'flood' events within the input datasets that were assigned a type of 'flash flood'; the perceptions of a small group of decision makers, potentially not representing the interests of the most vulnerable populations; and an overfitted representation of extreme precipitation risk rather than for risk of flash flood occurrence and associated impact on lives and livelihoods.

### 4.3. Limitations

The approach outlined here supports the use of designing early warning and humanitarian anticipatory action programming in Ecuador, however, other potential applications should be considered carefully. More specifically, the weighted mean approach for each indicator (described in Section 2.3.1) should be evaluated for fitness of use before being applied to other contexts. We suggest that increased attention be given to establishing

guidelines that specify the weighting of the flash flood geographical and description-derived confidence factors. Further, challenges remain related to the evaluation of the FFCI and eFFCI. The FFCI and eFFCI approaches were developed to build historical flash flood datasets when they are not available. However, with the intention to increase confidence in our approach, we compiled online media and newspaper reports on flash flood events to compare to the FFCI results in the study area. The flash flood content retrieved from the news media are reports for a specific day at one specific administrative level. The same specific day and administrative level can correspond to multiple flood records in our dataset, as the dataset is produced at a higher spatial resolution, with flooding indicated at latitude-longitude coordinates. From our dataset, we extracted all flood entries matching the specific day and administrative level mentioned in the media report, and estimated the fraction of those entries with an eFFCI > 5. Six of the nine flash flood-related media reports that we collected in Ecuador had corresponding entries in our dataset associated with that event in that district and day. As seen in Figure 7, all of the 6 corresponding media reports have at least one matching flash-flood entry in our database with eFFCI > 5, with 5 of the events having more than 60% of the matching flood entries with eFFCI > 5.

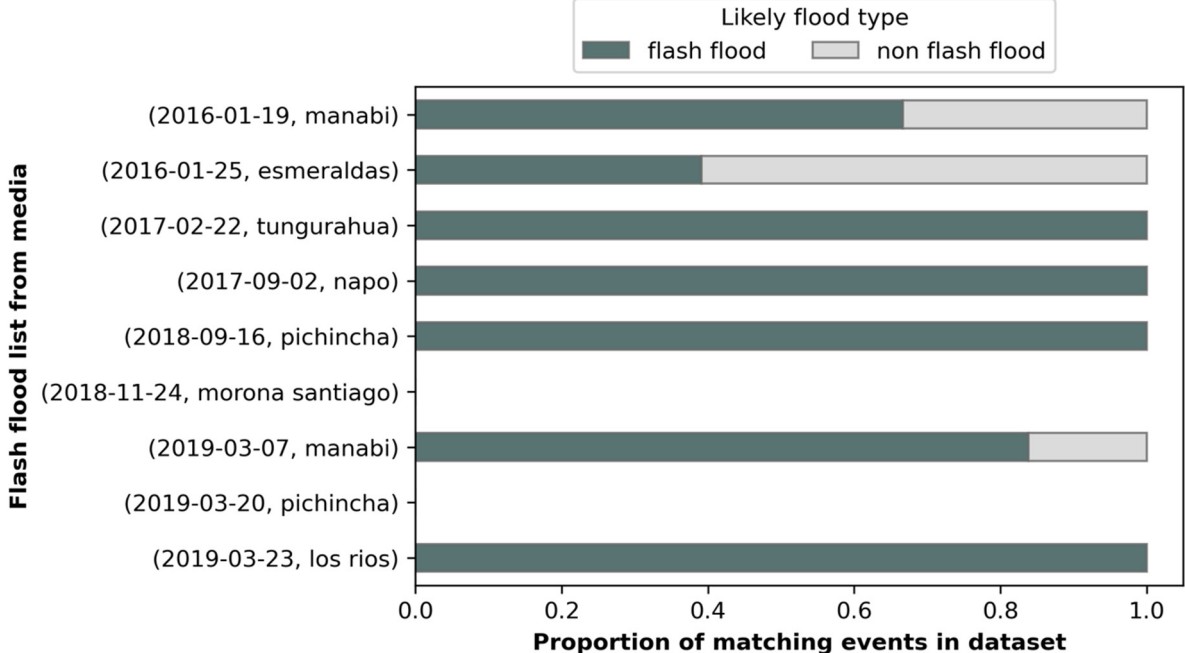

**Figure 7.** Proportion of likely flash flood events (eFFCI > 5) versus non likely flash flood events in our flood database matching the historical flash flood list extracted from media.

### 4.4. Recommendations

We recommend that resources are allocated for more precise reporting of all flood events, but most importantly for flash floods. We acknowledge that this will require a review of standard operating procedures, and likely additional capacity building, for disaster managers, the media, and (due to the increased importance of social media in disaster reporting) the general public, to characterize flood by subtypes and with to include details on impact. More type-specific data will improve flash flood model development, forecast production, early-warning systems and resilience programs [168]. There is a need for a coordinated effort to document and detect flash flood events through satellite remote sensing to improve the understanding of risk. Subsequently, we should consider the extent to which future satellite missions and research programs can be designed to inform specific applications within the disaster management and humanitarian communities. Further progress should continue, and ramp up, in developing satellite-data driven methods not only to monitor flash floods and other sudden onset and/or short-duration events,

but also to aid in the characterization, assessment of risk and forecasting of flood and disaster subtypes.

## 5. Conclusions

There is a need to enhance risk-assessment protocols for flash floods, as well as risk management and reduction activities including the development of standard operating procedures for early warning and anticipatory action, and long-term resilience strategies [167,169,170]. However, in comparison to other flood types, data required to enable policy and climate service development, such as timing, duration, location and impact level, is largely absent on a global scale for flash floods [171,172]. In particular, in many areas where an enhanced flash flood risk and vulnerability assessment could be of significant value, the necessary in situ data is more likely to be sparse [173,174]. Using information on non-flash floods (such as information on riverine floods) to drive flash flood specific applications, such as flash flood risk modelling and flash flood disaster management, could lead to significant biases that will further propagate through the subsequent elements of the application, such as the development of policy and services related to flash flood risk reduction and resilience including developing flood-type specific financial instruments and insurance mechanisms [175,176]. Here, we demonstrate that flood events can be assigned the subtype of flash flood with different levels of confidence. The approach is designed in acknowledgment of the differences in user type across use cases, thus, flexibility is built in, allowing for local and subject matter experts to identify appropriate thresholds of confidence related to flash flood likelihood. Those with various types of domain specific expertise, such as demonstrated here with the humanitarian community, should be involved throughout all steps of this process. The methods outlined here could add significant value to disaster risk management related to flash flooding if integrated within a cross-disciplinary approach, and if aligned with disaster management and climate service governance structures.

**Author Contributions:** Conceptualization, A.K. and A.B.; Data curation: A.B. and F.A.; Formal analysis: A.B.; Funding acquisition, A.K.; Methodology, A.K., A.B. and C.H.; Investigation, A.K., A.B. and C.H.; Writing—original draft preparation, A.K., A.B. and C.H.; Writing—review and editing, A.K., A.B., C.H., F.A., H.V., S.M., J.B. and A.d.S.; Supervision, A.K., H.V. and S.M.; Project administration, A.K. All authors have read and agreed to the published version of the manuscript.

**Funding:** This research was funded by the National Aeronautics and Space Administration, grant number 80NSSC18K0342 and 80NSSC18K1693.

**Data Availability Statement:** The two datasets resulting from this Ecuador Case Study are published on Zenodo. The historical dataset of flood events and impact with Flash Flood Confidence Indexes is not openly available. It has a restricted access because it contains data that are the property of the Ecuadorian National Secretariat for Disaster (SNGRE). The reference to the most updated version of the dataset is stated hereafter. *Bucherie, A., Ayala, F., Kruczkiewicz, A., 2021.: Ecuador historical flood occurrences and impacts dataset with Flash Flood Confidence Index (2007–2020), [Data set], Zenodo, http://doi.org/10.5281/zenodo.4662886, 2021.* The final Flash Flood Susceptibility Index of Ecuador catchments, based on geomorphology analysis is openly available as a shapefile. The reference to the most updated version of the dataset is stated hereafter. *Bucherie, A., and Kruczkiewicz, A., 2021: Ecuador Flash Flood Susceptibility Index (FFSI) based on catchment hypsometry, drainage and surface characteristics, [Data set], Zenodo, http://doi.org/10.5281/zenodo.4723395, 2021.*

**Acknowledgments:** We thank the Cruz Roja Ecuatoriana (Ecuadorian Red Cross), Servicio Nacional de Gestión de Riesgos y Emergencias (SNGRE) (The National Risk and Emergency Management Service of Ecuador) and Instituto Nacional de Meteorología e Hydrología de Ecuador (INAMHI) (The National Institute for Meteorology and Hydrology Service of Ecuador). This data supports the NASA GEO project: Towards A Global Flood & Flash Flood Early Warning Early Action System Driven by NASA Earth Observations and Hydrologic Models.

**Conflicts of Interest:** The authors declare no conflict of interest. The funders had no role in the design of the study; in the collection, analyses, or interpretation of data; in the writing of the manuscript, or in the decision to publish the results.

## Appendix A

**Table A1.** Discrete classification of Copernicus Land Use Land Cover (LULC) classes related to increased flash flood susceptibility.

| Copernicus LULC Discrete Classes | Classification (Values Increase with Increased Flash Flood Susceptibility) |
|---|---|
| Closed forest | 1 |
| Open forest | 2 |
| Snow and ice | 3 |
| Shrubs | 3 |
| Moss and lichen | 3 |
| Herbaceous wetland | 4 |
| Herbaceous vegetation | 5 |
| CroplandCropland | 6 |
| Bare/sparse vegetation | 7 |
| Urban/Built-up | 8 |

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
