# Peer review of "Development of a Flash Flood Confidence Index from Disaster Reports and Geophysical Susceptibility"

_remotesensing, doi:10.3390/rs13142764_

Round 1
Reviewer 1 Report
I have doubts about the method that is proposed to identify flash floods. How are the classes and subclasses derived? How are scores of each subclass determined? A lot seems to be based on expert judgement. Would a different choice for these subclasses and scores lead to a different result, i.e. a different set of identified flash floods? Line 167: Next, four classes are defined … > I think the term class is misleading. Classes are usually mutually exclusive. Would property or attribute not be a better term? Table 1: classess and subclasses > This choice of classes and subclasses does not correspond with the definition of a flash flood on Line 40: “rapid onset hydrological events triggered by localized, intense rainfall events, resulting in unexpected, sudden and high velocity flow occurring in small rivers or ephemeral streams, as well as surface runoff”. So how do you arrive at these four classes and ten subclasses? Line 161: We first define a list of flash flood subtype ‘identifiers’ as terms and phrases more frequently present in flash flood reports compared to other flood subtypes. > How did you select these identifiers? How often do they occur in the reports for Equador? Table 2: LULC directly impacts runoff generation and behavior [98,129,131]. Discrete LULC classes are reclassified into flash flood susceptibility scores from 1 to 8, depending on potential to influence surface runoff. Closed forest = score of 1, urban environment = score of 8. > It would be interesting to know how your ranked the other classes Figure 2 > What do you mean by mean Strahler order, LULC and sand content? Mean over catchment area? Line 350 and further> I have difficulty understanding what you mean here. Be more concrete. How do you envisage that the FFCS is used? Line 373: how do you see this being used in early warning or risk communication? In do not agree with the benefit of FFCI as described on Line 385 and further. The FFCI identifies flash flood events accoring to the definition of the FFCI. You could set up a different flash flood classification and scoring method and end up with a different selection of events. In this different setup, you would still have identified the FFCI events from the larger set. The selection is only useful if the selection is validated against an independent source, for example eye witnesses or impact figures. Line 403: I fully agree: The challenge remains to validate the FFCI and eFFCI. Line 405: Six out out of nine events confirmed? That is a very small number of events for a validation study. What about the three events that were not confirmed? Does this mean the FFCI has a 35% miss rate? Line 408: Do you mean 60% of the six matching flood events? Was that 3 or 4 events?
Author Response
Dear Reviewer 1
We thank you for your time in reading and reviewing our paper. Thank you for your comment throughout. We have reconstructed elements of the paper based on your feedback. We have additional information in the response attached.
Best regards,

Reviewer 2 Report
The paper presents very nicely the work that has been achieved for the development of the index and its application in Ecuador. The issue of flash flood prone areas identification is a major concern around the world and the proposed method is providing a significant result that can support the approach. A minor comment would be to mention clearly the limits of the proposed approach and the associated uncertainties both for the observation data and the collected field damages reports.
Author Response
Dear Reviewer 2
We thank you for your time in reading and reviewing our paper. Thank you for your comment throughout. We have reconstructed elements of the paper based on your feedback. We have additional information in the response attached.
Best regards,

Reviewer 3 Report
I attach a file.

Author Response
Dear Reviewer 3
We thank you for your time in reading and reviewing our paper. Thank you for your comment throughout. We have reconstructed elements of the paper based on your feedback. We have additional information in the response attached.
Best regards,

Round 2
Reviewer 1 Report
The manuscript has definitely improved. I still have doubts whether the method is applicable generally. But it should be of interest to the flash flood community.